# Symbolic metaprogram search improves learning efficiency and explains rule learning in humans

Joshua S. Rule [1] ✉, Steven T. Piantadosi [1], Andrew Cropper[2], Kevin Ellis [3], Maxwell Nye[4] & Joshua B. Tenenbaum [5]

Throughout their lives, humans seem to learn a variety of rules for things like applying category labels, following procedures, and explaining causal relationships. These rules are often algorithmically rich but are nonetheless acquired with minimal data and computation. Symbolic models based on program learning successfully explain rule-learning in many domains, but performance degrades quickly as program complexity increases. It remains unclear how to scale symbolic rule-learning methods to model human performance in challenging domains. Here we show that symbolic search over the space of metaprograms—programs that revise programs—dramatically improves learning efficiency. On a behavioral benchmark of 100 algorithmically rich rules, this approach fits human learning more accurately than alternative models while also using orders of magnitude less search. The computation required to match median human performance is consistent with conservative estimates of human thinking time. Our results suggest that metaprogram-like representations may help human learners to efficiently acquire rules.

Humans acquire a wide variety of concepts throughout their lives, many of which are well-described as rules, i.e. symbolic expressions in a kind of mental language or language of thought[1]. Category learning, for example, can be described as learning a rule which accepts or rejects potential category members based on their individual features[2–5]. Similarly, procedure learning can be described as acquiring a rule for which behaviors to sequence together in what order[6–8]. Theory learning can also be described as acquiring a network of rules explaining the relationships between various causes and effects[9–11]. The exact scope of human rule-learning is unclear: even if they can describe a wide variety of concepts[12,13], theories of rule-learning face a number of challenges[14–16]. Moreover, exactly how many concepts are actually represented using rules is an often difficult empirical question, as seen, e.g., in debates over how humans process past-tense constructions in English[17,18]. Even so, rules are a significant part of humans' cognitive landscape.

Moreover, many of the rules people learn are algorithmically rich. They go beyond associative pairings or even simple logical or arithmetic formulae to encode a series of steps with a variety of algorithmic content[19]. For example, the rules children learn for basic arithmetic require pattern matching, conditional reasoning, iteration, recursion, maintaining state, and caching partial results. Beyond logic and mathematics, these sorts of complex rules appear in domains as varied as game playing, social reasoning, food preparation, and natural language understanding.

Theories of how people acquire algorithmically rich rules must not only explain task performance but must also capture other hallmarks of human learning. While there are many, we focus here on three. First, the representations should be interpretable in ways that support the kinds of composition, explanation, sharing, and reuse we see in humans[1,20–23]. Second, learning should also be possible from sparse data on the scale

[1]Psychology, University of California, Berkeley, Berkeley, CA 94704, USA. [2]Computer Science, University of Oxford, Oxford, UK. [3]Computer Science, Cornell University, Ithaca, NY 14850, USA. [4]Adept AI Labs, San Francisco, CA 94110, USA. [5]Brain and Cognitive Sciences, Massachusetts Institute of Technology, Cambridge, MA 02139, USA. ✉e-mail: rule@berkeley.edu

that people realistically encounter[24,25]. Third, learning should require only moderate amounts of computation and search, consistent with human limits on thinking time and cognitive resources[25,26].

One theory of rule-learning treats the language of thought as a sort of mental programming language, such that learning proceeds by constructing program-like representations. For example, the concept LIFT could be a simple program combining primitives for CAUSE, GO, and UP to mean, roughly, "cause to go up"[27]. This approach makes human learning analogous[12] to program induction[28]—discovering programs to explain data. Humans learning new rules, much like computer programmers writing new programs, fluidly operate over a broad space of computations and appear to efficiently construct interpretable structures from sparse data[19]. Symbolic programs provide interpretable hypotheses by decomposing complex computations into discrete and semantically meaningful parts—i.e. simpler computations—that support modular explanation, reuse, and sharing[29]. Program-induction models are also typically data efficient, learning from relatively few observations. Human learning has been modeled as program induction in many domains, including structure discovery[30], number acquisition[31], rule learning[32], physical reasoning[33], memory[34], and cultural transmission[35]. They have even been applied in domains seemingly resistant to program-based approaches, such as perceptual learning[36–39], language learning[40–42], and motor learning[43,44].

Despite successes, program induction models face a fundamental obstacle: the hard problem of search. The space of possible programs grows exponentially in both program length and the number of primitive operators; it is unclear how to narrow the search space to prevent combinatorial explosions[45]. While continuous weights and differentiable error functions scale gradient-based search to arbitrarily complex neural networks[46], no effective methods exist for the highly discontinuous spaces of symbolic programs. The need for effective search mechanisms is so intense that it has been hypothesized as a motivating force behind play[47] and childhood[48], highlighting just how significant it is that program induction models lack this ability.

To help address this problem, this paper focuses on a hypothesis about a class of representations which might help people search efficiently over program-like content. More specifically, we hypothesize that in addition to object-level content, people directly incorporate sophisticated forms of reasoning into their hypotheses. We predict that doing so reshapes inductive biases by simplifying relevant hypotheses[49] and making them easier to find.

This hypothesis does not fit cleanly into the classic Marr levels[50]. It makes a theoretical claim not about a general computational problem or specific representation but instead about a class of representations, i.e. something between a computational and algorithmic-level claim. While many algorithmic-level details, such as the specific search algorithm, the particular domain, and even the content of individual metaprimitives, are significantly less important to our claims, we assume that algorithmic concept learning does involve a serial search process that cannot involve too many steps. These are algorithmic-level constraints on human thinking and we seek an algorithm that is consistent with them.

We therefore instantiate a version of this hypothesis in a model called MPL (MetaProgram Learner), which incorporates metaprograms—programs that revise programs—into its representation language. We test MPL against humans alongside recent and classic baselines on a benchmark of 100 program induction problems.

Before describing MPL, we present the task domain and outline our benchmark. The domain consists of list functions[51–54], where learners encounter datasets pairing input and output lists of numbers. To see how learning in this domain might resemble program induction, consider $\mathcal{F}$, a list function where:

$$[1, 3, 9, 7] \xrightarrow{\mathcal{F}} [1, 1, 3, 3, 9, 9, 7, 7] \tag{1}$$

Brief observation leads most people to a strong hypothesis. They notice that values in the output appear twice consecutively, suggesting duplication. Each input element also appears in the output in the same order. Together, these features suggest an iterative process like: repeat every element two times in order of appearance. This rule seemingly has no strong competitors, a sense that grows after seeing more examples:

$$[1, 3, 9, 7] \xrightarrow{\mathcal{F}} [1, 1, 3, 3, 9, 9, 7, 7] \tag{2}$$

$$[6, 9, 2, 8, 0, 5] \xrightarrow{\mathcal{F}} [6, 6, 9, 9, 2, 2, 8, 8, 0, 0, 5, 5] \tag{3}$$

$$[9, 2] \xrightarrow{\mathcal{F}} [9, 9, 2, 2] \tag{4}$$

People see up to eleven examples in our experiments, but nearly all participants acquire this rule within three examples. Program induction models might hypothesize that learners represent it with a program like:

$$\mathcal{F} = (\lambda \; \texttt{xs} \; (\texttt{if} \; (\texttt{empty xs}) \; \texttt{xs} \; [(\texttt{head xs}), (\texttt{head xs}) | (\mathcal{F}(\texttt{tail xs}))])) \tag{5}$$

$(\lambda \; \texttt{xs} \; \ldots)$ uses the $\lambda$ operator from $\lambda$-calculus, which here creates a function taking a list, xs, as input. $(\texttt{if} \; (\texttt{empty xs}) \; \ldots)$ tests whether xs is empty. If so, $\mathcal{F}$ returns xs; there is nothing to duplicate. Otherwise, $[(\texttt{head xs}), (\texttt{head xs}) | \ldots]$ creates a list repeating xs' first element, or head, twice ($[\texttt{x}, \ldots | \texttt{zs}]$ prepends $\texttt{x}, \ldots$ to the list zs). $(\mathcal{F}(\texttt{tail xs}))$ completes the list by recursively applying $\mathcal{F}$ to xs' remaining items, or tail.

Some list functions are harder to learn. Consider $\mathcal{G}$:

$$[7, 9, 0, 2, 6, 8, 3, 4, 6] \xrightarrow{\mathcal{G}} [0, 9, 7, 4, 6, 3] \tag{6}$$

Some people may notice that the output contains a subset of the input elements, but there seems to be no obvious pattern. Unlike with $\mathcal{F}$, it is difficult to form strong hypotheses without more data:

$$[7, 9, 0, 2, 6, 8, 3, 4, 6] \xrightarrow{\mathcal{G}} [0, 9, 7, 4, 6, 3] \tag{7}$$

$$[1, 7, 8, 2, 5, 6, 1] \xrightarrow{\mathcal{G}} [8, 7, 1, 4, 5, 1] \tag{8}$$

$$[6, 7, 1, 3, 2, 0, 8, 9, 4, 5] \xrightarrow{\mathcal{G}} [1, 7, 6, 4, 2, 8] \tag{9}$$

Many people remain puzzled even after studying these examples. About half of our participants never acquire a rule for $\mathcal{G}$; the others usually need three to five examples. Those who do acquire it may notice several unlikely coincidences. First, $\mathcal{G}$ does not trivially map every input to the same output. Second, input length varies but the output always has six elements. Third, many but not all input elements appear in the output (perhaps $\mathcal{G}$ filters elements using some test or shuffling operator). Fourth, shared elements differ in order, so filtering seems unlikely. Fifth, fixed positions in the input are copied to fixed positions in the output. Element 1 becomes element 3, 2 stays 2, 3 becomes 1, 5 stays 5, and 7 becomes 6. Finally, output element 4 is always 4.

Each observation identifies a simple pattern produced by aligning shared structure in the data. Putting them together leads to the rule: elements 3, 2, 1, the number 4, then elements 5 and 7. While this rule explains the data, it seems unusual. We can nevertheless model it as

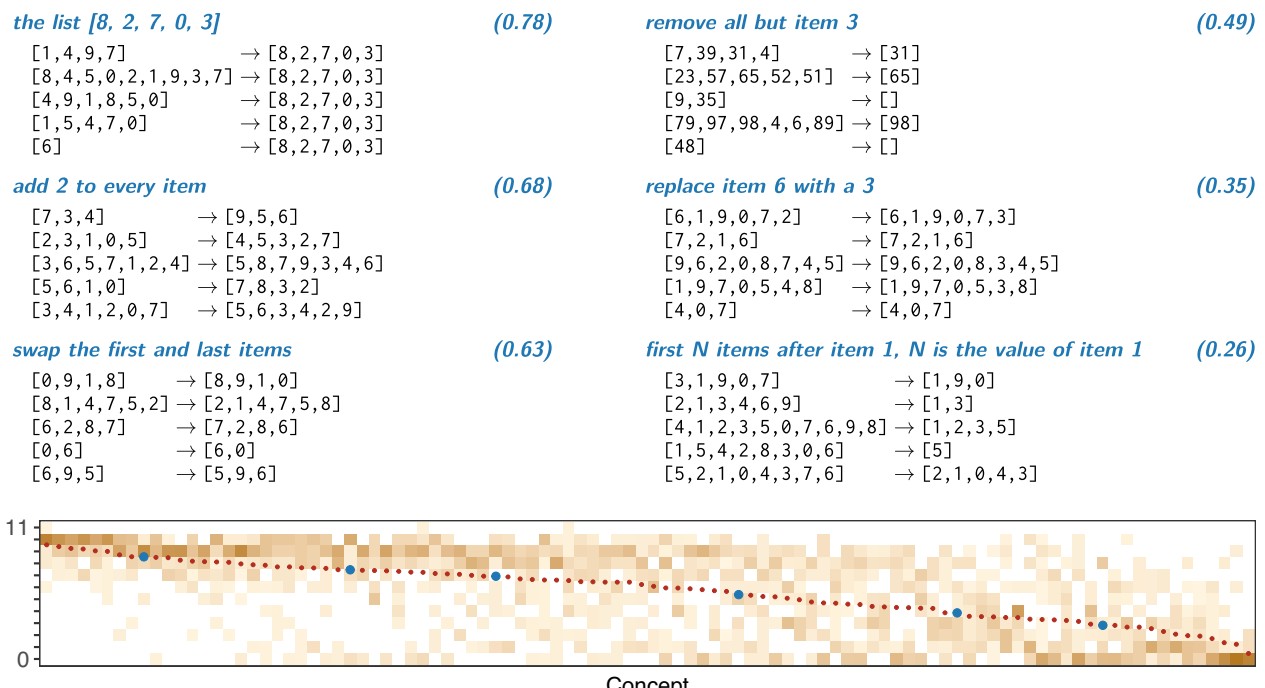

**Fig. 1 | List functions vary widely in difficulty and algorithmic content.** Six list functions with an English description, human mean accuracy ($n = 389$ people) in parentheses, and input → output examples. Plot shows empirical distribution over accuracy per function (100 functions) for humans (darker = more mass); dots show mean accuracy with example functions marked in blue.

the program:

$$\mathcal{G} = (\lambda\ \mathtt{xs}\ (\mathtt{swap}\ 3\ 1\ (\mathtt{replace}\ 4\ 4\ (\mathtt{cut}\ 6\ (\mathtt{take}\ 7\ \mathtt{xs}))))) \tag{10}$$

It again uses $\lambda$ to create a function binding xs, ($\lambda\ \mathtt{xs}\ \ldots$). Working from the inside out in the remaining expression, ($\mathtt{take}\ 7\ \mathtt{xs}$) takes the first seven elements, ($\mathtt{cut}\ 6\ \ldots$) removes the sixth, ($\mathtt{replace}\ 4\ 4\ \ldots$) replaces the fourth with a 4, and ($\mathtt{swap}\ 3\ 1\ \ldots$) swaps the first and third. Composing a few simple operations represents an unlikely concept that can still be learned from sparse data.

Like other classic domains such as numerical functions[55–58] and Boolean functions[2,3,5,49,59], list functions might superficially seem abstract and focused on a narrow corner of human cognition, but they are well suited to empirical study and modeling of how people learn rules. Numbers and sequences both have a long and productive history in the study of human learning[8,37,60–63]. List functions are in fact particularly useful for testing the sorts of program-learning models of concept learning which have now been deployed to explain rule-learning in dozens of domains[11,12,19,27,30–44]. They provide a general and well-controlled setting where problems vary widely in difficulty and algorithmic content (the domain is Turing-universal) and can be tested easily in humans and machines. Indeed, many bear a strong resemblance to everyday tasks such as sifting out junk mail (filtering); counting the books on a shelf (folding separate items into a composite result); alphabetizing a list of names (sorting by a criteria); and decorating a tray of cupcakes (mapping a transformation over a collection of items). Being analogous, however, does not mean that we claim that the tasks are equivalent. In more naturalistic cases, it seems likely that context-specific knowledge effects[64] aid learning, but our results (including the replication described in Supplemental Note 8) show that in this domain, as in many others, people can rapidly acquire and apply rules from sparse data. Human performance on our task in particular is far above chance and remains interesting in its own right.

We conducted a study of human and machine concept learning by constructing a benchmark of 100 list functions that vary widely in learnability (Fig. 1). The set includes $\mathcal{F}$ and $\mathcal{G}$, so the discussion above is relevant to the entire benchmark. Our primary goals in constructing this benchmark were to collect functions: demonstrating broad variation in learning difficulty for humans (i.e. not dominated by floor/ceiling effects); which could be described with a small set of primitives; and that are easy enough to learn that the performance of program induction models would not be dominated by floor/ceiling effects. Moreover, testing our hypothesis requires problems where we can compare solutions which do and do not incorporate representations of structured reasoning. Most benchmark problems thus emphasize reasoning techniques which MPL can leverage during search. We compared MPL's performance on this benchmark to leading alternative explanations of human rule-learning.

Building on the idea of learning over program-like representations, our approach to concept learning draws on three core insights inspired by the techniques of human programmers[19].

First, most program learning models search over programs composed of object-level primitives, such as $\mathtt{head}$ and $\mathtt{take}$ in Eqs. (5) & (10). Assuming search operators are fully parameterized, programs can also be described using the decisions required to produce them during search. These decisions describe how to construct a program, namely by repeating the search process producing it. While this process is typically implicit in search algorithms, programmers often consider it explicitly, discussing transformations and their effects—e.g. swapping iteration for recursion or extracting repeated code into a shared function—in addition to actual code.

Second, many search algorithms apply a single generic operator, e.g. enumerating from a grammar or sampling from a distribution. Some bias search toward the best hypotheses discovered so far: consider Markov chain Monte Carlo's accept/reject step[65]; particle filtering's resampling[66]; or genetic programming's tournaments[67]. Even so, learning inefficiently relies on accumulating small, often random, local changes. By contrast, human programmers can flexibly combine hundreds of structured techniques for revising programs[68]. Many cater to

particular problems and specify context-dependent solutions, much like high-level actions in hierarchical planning[69].

Third, many search algorithms begin without regard for available data, e.g. starting from the lexicographically first program or a random sample. Such hypothesis-driven learning generates proposals independently of data[70]. These methods are very general but must discover relevant structure by chance rather than by inferring it from data. By contrast, data-driven learning generalizes input/output pairs directly into a program using some inference technique, e.g. for detecting recurrent structure. It minimizes search but requires strong assumptions that sharply constrain which programs can be learned from what data. Human programming techniques supersede both approaches in many ways: they are often designed to expose latent structure and can be flexibly composed to apply to nearly any problem. They can thus be rooted directly in the data whose structure needs to be explained (e.g. "recursive data can be rewritten like so" or "if data contains repetition with minor differences, perhaps those differences can be abstracted away.").

Given these observations, we hypothesize that people extend languages of object-level primitives with patterns of structured transformation called metaprimitives. Some metaprimitives might simplify repeated structure; others might memorize data for further analysis or to encode exceptions. On this view, primitives and metaprimitives can be freely composed into expressions called metaprograms that combine object-level content and structured transformation. Metaprimitives operate on structures built of primitives, so a metaprogram can always be evaluated to produce a program without metaprimitives. That is, metaprimitives provide an alternative way of expressing certain programs, shifting the inductive bias so that they become easier to describe.

While the introduction of new object-level primitives also shifts the inductive bias[49,71], metaprimitives can capture different kinds of bias from object-level primitives. In particular, object-level primitives cannot leverage the internal structure of their arguments; they must treat those arguments as black boxes. By contrast, metaprimitives are program transformations and so can change their behavior based on this internal structure. For example, the MPL model includes a metaprimitive called `AntiUnify`, whose primary effect is to introduce variables into programs. There are many ways to do this, and considering them all would require a long search. `AntiUnify`, however, uses the structure of the input program to decide where to introduce variables without additional search. That is, `AntiUnify` uses the structure of its arguments to ignore portions of the search space which methods using only object-level primitives would otherwise have to consider.

Metaprimitives thus take advantage of all three insights above. First, they make program transformations an explicit part of the language instead of leaving them only implicitly available as search operators. Second, just as languages typically contain many primitives, they can also contain many metaprimitives, each expressing a different program manipulation. Third, if some metaprimitives can memorize data, other metaprimitives can extract information from those data and learn more efficiently than using primitives alone by introducing different kinds of inductive bias. By encoding search operators reminiscent of data-driven search and embedding them into the language of a hypothesis-driven learner, metaprimitives perhaps combine the best of both approaches.

To evaluate these ideas, we implement MPL, a symbolic learner which extends traditional program induction approaches by incorporating metaprimitives. We seek to investigate the usefulness of the metaprimitive approach rather than to make strong claims about any specific metaprimitive. The particular metaprimitives implemented here (Table 1; Supplementary Note 2) thus capture relatively simple patterns of reasoning inspired by operators in inductive logic programming[72], analytical induction[73], automated theorem proving[74],

and refactoring techniques in software engineering[68]. In practice, some metaprimitives do more work than others but each describes an operation for reasoning about program structure.

Program-induction-based models of concept learning often use languages whose primitives (and in this case, metaprimitives) are closely related to the concepts being studied. This can be seen, for example, in recent work on learning in the domains of number[31,75], logic[49,76], and geometry[37,77], among others. The claim is not that these limited languages constitute a learner's entire mental repertoire, nor that the studied domain is the only one in which humans are capable of learning. Nor is the claim that the simple existence of computational primitives (or metaprimitives) is enough to explain human learning, or that any existing model is sufficient to explain all of human learning. They are instead case studies comparing a plausible set of primitives and learning dynamics against human learners in a particular domain. We take the same approach in introducing metaprimitives.

Metaprimitives are useful for working with list functions because they capture patterns of reasoning (e.g. simple forms of structure mapping, composition, generalization) that are useful for reasoning about lists specifically or about programs generally, similar to human code manipulation techniques. Previous learning systems embed these operators directly into search algorithms and apply them in stereotypical patterns. Explicit metaprimitives allow MPL significantly more flexibility than previous models.

Figure 2A–C illustrates MPL using $\mathcal{F}$, described earlier. Given examples (Fig. 2A), MPL learns a metaprogram (Fig. 2B) combining primitives—namely the empty program, $\varepsilon$—and metaprimitives. `MemorizeAll` adds data directly to a program, making their latent structure available to other metaprimitives. `Recurse` hypothesizes that rules involving certain limited transformations of linearly recursive structures (e.g. elementwise transformations of lists, unary numbers, strings) can themselves be recursively decomposed into simpler rules. Here, it captures people's observation that each input element explains two consecutive output elements by aligning and unrolling input/output lists. This change reveals latent structure but introduces many new rules. `AntiUnify` is helpful here. It uses anti-unification—an important program synthesis technique[78,79]—to compute a least-general generalization that systematically aligns shared structure across rules into a single general rule. For example, comparing `F[1|[3, 9, 7]] ≈ [1, 1|(F[3, 9, 7])]` and `F[3|[9, 7]] ≈ [3, 3|(F[9, 7])]` reveals a common structure: the first element is repeated twice, and the rest of the list is processed recursively. `AntiUnify` discovers a corresponding rule, `F [x | y] ≈ [x, x | (F y)]`, by similarly aligning common structure and generalizing over differences.

Because metaprimitives represent program transformations, applying a series of metaprimitives produces intermediate results and then a final program that both explains the data and can be applied to novel inputs (Fig. 2C). Because MPL can freely mix primitives and metaprimitives, it can also learn programs directly, e.g. for problems where available metaprimitives are not applicable.

Figure 2D–F repeat the process for $\mathcal{G}$. While $\mathcal{G}$ is complex to describe in English, its metaprogram is even simpler than $\mathcal{F}$'s. Lacking recursive structure, $\mathcal{G}$ can be described using structural alignment alone. After encoding data with `MemorizeAll`, a call to `AntiUnify` is sufficient. The resulting program, however, is more complex than the one for $\mathcal{F}$. MPL is sensitive to this complexity, which helps to explain why $\mathcal{G}$ is harder to learn than $\mathcal{F}$. While the metaprogram is simple, the complexity of the resulting program requires observing a sufficient amount of data.

To balance simplicity and fit, MPL models learning as MAP inference in a Bayesian posterior over metaprograms. Computing the posterior exactly is intractable; MPL approximates it using Markov Chain Monte Carlo (MCMC) over programs[42,76] extended to the space of metaprograms. Monte Carlo methods are notable as rational

**Table 1 | MPL relies on primitives and metaprimitives**

| MPL Metaprimitives | | |
|---|---|---|
| **Kind** | **Usage** | **Description** |
| ● | `(MemorizeAll p)` | add all data to program p |
| ● | `(Memorize p ψ)` | add datum ψ to program p |
| ◆ | `(Recurse p ψ)` | add recursion ψ to program p |
| ◆ | `(Delete p ψ)` | delete rule ψ in program p |
| ◆ | `(Variable p ψ)` | add variable ψ to program p |
| ◆ | `(Compose p ψ)` | add composition ψ to program p |
| ◆ | `(Subproblem p ψ)` | extract problem ψ from program p |
| ◆ | `(AntiUnify p)` | unify similar rules in program p |
| **Object-Level Primitives** | | |
| ■ | ε | the empty program |
| ■ | `(λ x body)` | bind variable x for use in body |
| ■ | `0, 1, 2,…, 99` | natural numbers |
| ■ | `nan` | number < 0 or > 99 |
| ■ | `true, false` | Boolean values |
| ■ | `[]` | empty list |
| ■ | `[x|xs]` | prepend x to xs |
| ■ | `(+ x y)` | add x and y |
| ■ | `(- x y)` | subtract y from x |
| ■ | `(> x y)` | true if x is less than y |
| ■ | `(if p a b)` | a if p is true, else b |
| ■ | `(== x y)` | true if x and y are identical |
| ■ | `(is_empty xs)` | true if xs is empty |
| ■ | `(head xs)` | first element of xs |
| ■ | `(tail xs)` | drop the first element of xs |
| ■ | `(fix x f)` | recursively apply f to x |

MPL used metaprimitives for observation (orange circles) and inference (green diamonds); ψ represents a random choice. All models also used object-level primitives (blue squares).

process models[80], addressing computational-level concerns with psychologically plausible methods. This approach might appear to suffer from the problem that we identified earlier of learning inefficiently via small, local changes. Searching over metaprograms, however, helps to address this problem. Because metaprimitives can encode arbitrary program transformations, even small changes can have large, nonlocal impacts on the resulting program.

## Results

We compare MPL to a variety of symbolic, neural, and neurosymbolic models of learning, namely Fleet[42], Enumerate[71], Metagol[81], RobustFill[82], and Codex[83] (See Methods for additional motivation and details on each model). All models except Codex use similar primitives (Table 1) adapted to their computational paradigms (e.g. lambda calculus, first-order logic, term rewriting); Codex uses the python programming language. Only MPL uses metaprimitives to construct metaprograms, which comprise its central hypothesis. Critically, these metaprimitives represent structured ways of manipulating the primitives; they change the inductive bias, but not the theoretical expressiveness of MPL. Given enough time, each model will find a solution if it exists. The critical questions are then how quickly solutions can be found and whether adding metaprimitives to the representation language's compositional basis improves the speed with which high-quality solutions are found.

This paper evaluates metaprimitives as an explanation of how humans rapidly acquire complex rules. We therefore focus on the rate of acquisition, considering a rule acquired on trial $n$ if the learner gives correct responses on all trials ≥$n$. In these experiments, participants complete a trial by observing an input list, typing in and submitting a predicted output, and then observing the correct output. Because perfect performance is a strict test of learning, we also examine mean accuracy. On these measures, human list function learning provides a challenging target for model learners (Supplementary Note 3). 54% of functions were acquired by ≥50% of human learners within eight trials. This value is high given that chance performance on any single trial is approximately 1 in $10^{30}$. 50% of functions were acquired by at least one person after a single trial, 75% after two trials, and fully 99% within eight trials. Only 2% were acquired by all participants within eight trials. Mean human accuracy tells a similar story. Averaging across functions, it was high (Mean = 0.521, 95% CI [0.479, 0.559]; SD = 0.202, 95% CI [0.180, 0.221]) relative to chance, and ranged from 0.042 to 0.868 for individual functions. Supplementary Note 8 reports similar results for a replication.

Participants' performance is perhaps particularly impressive given their relatively low levels of programming experience. Of the 392 participants in our sample, 259 (66%) provided an interpretable free-response statement of their prior programming experience. Of these, 151 (58%; mean accuracy = 0.49) indicate no prior programming experience, an additional 27 (10%; mean accuracy = 0.50) indicate social exposure to programming concepts and perhaps simple website construction. 43 (17%; mean accuracy = 0.50) report encountering programming through introductory coursework or by building several websites. Only 38 (15%, mean accuracy = 0.53) indicate significant academic or professional exposure to programming (See also Supplementary Note 8).

Figure 3A compares humans to models given a large search budget. Only MPL (500K) and Fleet (500K)—so named because each takes 500K search steps per trial—explain human behavior well in this

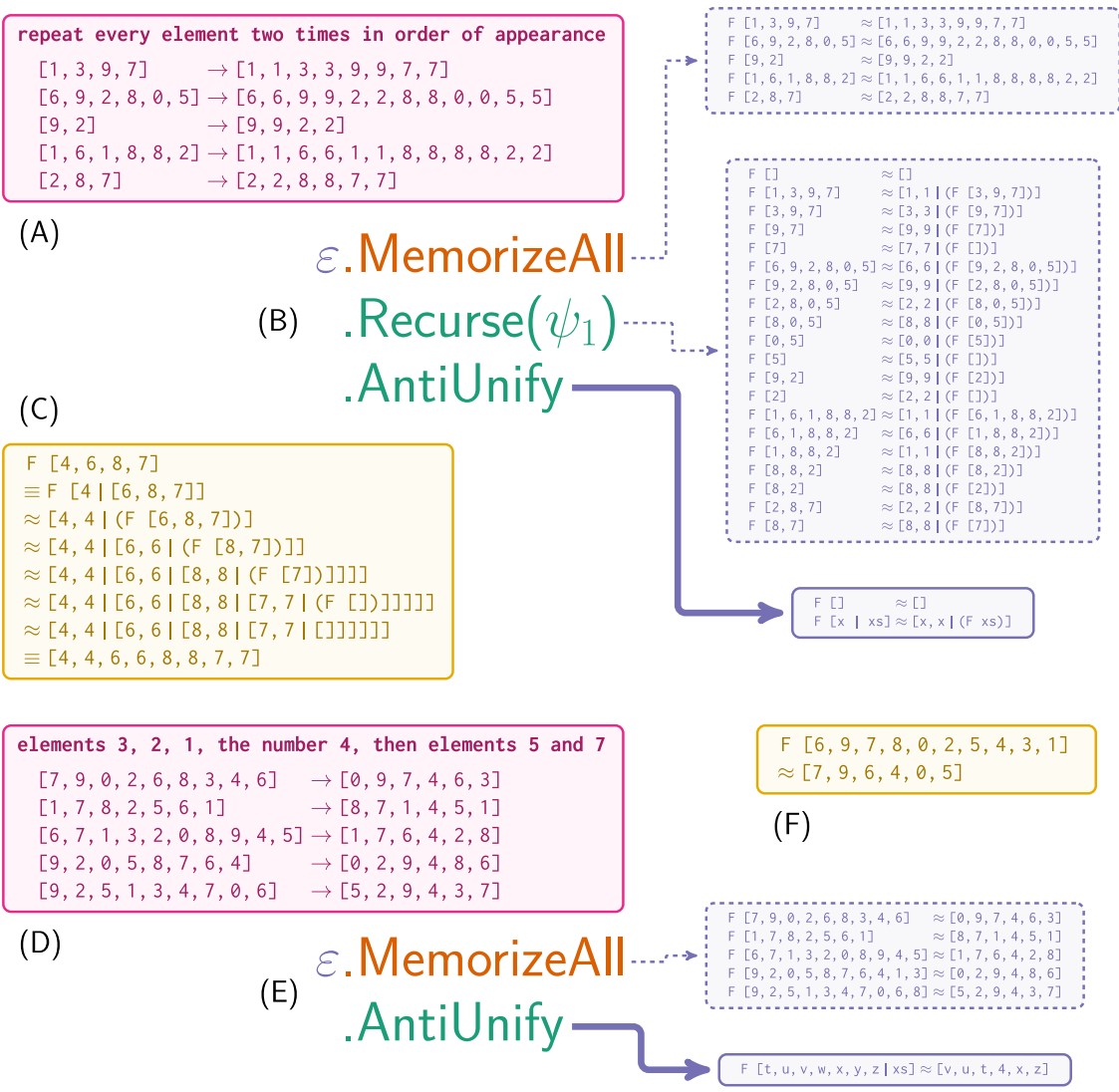

**Fig. 2 | Two examples of how MPL uses metaprograms to discover programs.**
**A** The target function (not observed by MPL) and observed input/output pairs.
**B** MPL searches over metaprograms which compose primitives (blue) and meta-primitives for observation (orange) and inference (green). **A, B** is shorthand for (**B**) (**A**). Given data, metaprograms can be reduced to programs of primitives (solid blue box), often via intermediate programs (dashed blue boxes). **F** represents the target function; [x, y,…, z | xs] is shorthand for prepending elements x, y,…, z to list xs; $\psi_i$ represents uniformly random selection among multiple options so that metaprograms reduce deterministically. **C** Applying the learned program to novel data. **D**–**F** A second example.

setting. Figure 3B compares model and human mean accuracy for each function; again, only MPL (500K) and Fleet (500K) capture human-level performance. By contrast, Enumerate, Metagol, and RobustFill failed to achieve human-level accuracy, performing at or below humans' 25th percentile and deviating significantly from human mean accuracy. Codex inhabits a middle ground, acquiring approximately as many functions as 25th percentile humans and similarly deviating from human mean accuracy.

Both Fleet and MPL implement MCMC over programs, a form of stochastic hillclimbing which probabilistically accepts new hypotheses—typically incremental updates to current hypotheses—based on their score relative to the current hypothesis. They thus encourage rapid improvement by generally accepting only small, beneficial changes. By contrast, both Enumerate and Metagol use exhaustive search algorithms. As target programs grow more complex, exponentially many simpler programs must be considered. Most functions in our dataset are simply too complex for them to discover even with tens of millions of search steps. RobustFill is neither exhaustive nor hillclimbing but generates independent samples (conditioned on the training data), which is extremely inefficient for low-probability programs. Codex also

generates conditionally independent samples, but its significantly larger training set and more sophisticated architecture help it to outperform RobustFill.

While MPL (500K) and Fleet (500K) both perform well, there are important differences between them. For example, both models fail to predict a single trial correctly for a small number of unique functions (MPL = 12, Fleet = 13). For Fleet, these include a mix of recursive and non-recursive problems primarily characterized by long description lengths. For MPL, none deal with non-recursive structural reasoning (e.g. indexing, swapping, removing elements). Metaprimitives like `Antiunify` and `Variablize` give MPL an advantage over Fleet on these problems. Instead, all twelve involve recursion. The `Recurse` metaprimitive captures a limited form of recursion (see Supplemental Note 2), and eleven of the twelve use recursive patterns for which MPL has no relevant metaprimitive. Without appropriate metaprimitives, solutions to these problems are difficult to discover. While humans struggle with some of these problems—using the first two elements of the input list to specify a sublist of the remaining elements has a mean human accuracy of just 4.2%—others like computing the maximum element, computing the sum of the elements, and reversing the

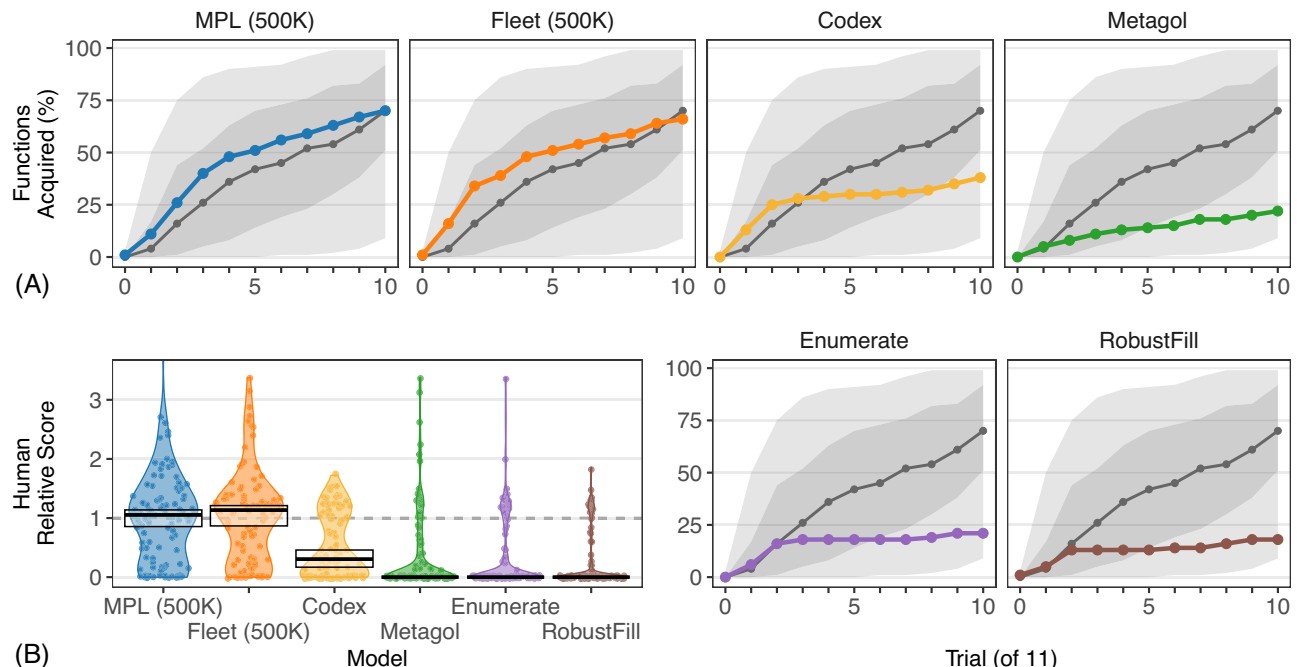

**Fig. 3 | MPL and Fleet outperform other models given large search budgets.**
**A** Percentage of functions (100 total) acquired per model (subplots) by a given trial (11 total) with human median performance ($n = 389$ people; gray curve), 25%-75% human performance (dark gray band), and best-worst human performance (light gray band). We measure acquisition using the strict criterion of generating correct predictions on all future trials. **B** Ratio of model mean accuracy to human mean accuracy ($n = 389$ people) per concept (dots; 100 total) per model, with parity between models and humans (dotted line) and a kernel density estimate (colored regions). The crossbars show the median across functions with a 95% bootstrapped CI. Each model is associated with a unique color for easier comparison across figures.

elements have human mean accuracies well above 50%. More generally, MPL is highly accurate in producing non-recursive solutions to non-recursive problems; MPL (500K) does so in 97.0% of runs. It is less accurate in producing recursive solutions for recursive problems; MPL (500K) does so for just 34.4% of runs.

Only Fleet (500K) and MPL (500K) match human performance while acquiring explainable hypotheses from sparse data. We now consider another important aspect of human learning: search efficiency. Human cognition is resource-constrained[25]; many forms of reasoning are well-modeled with just a handful of search steps[26]. MPL and Fleet differ in how well they approximate human behavior with more cognitively plausible resources. MPL learns much faster than Fleet given a fixed dataset. Each thin curve in Fig. 4A plots the posterior probability of the best hypothesis discovered by a given step as a result of search (i.e. not the posterior probability of the generating function, to which neither model ever had access) for either Fleet or MPL for one of the 100 functions. It also plots the mean of these scores when averaging across all 100 functions (thick curves). Because the two models were tested on the same functions and ultimately searched the same space of programs (i.e. MPL's metaprograms compile to programs in Fleet's search space), these curves demonstrate how efficiently the models search relative to one another. Notably, this mean posterior probability of the best discovered hypotheses is higher for MPL at five thousand search steps than for Fleet at five million, suggesting that MPL discovers concise descriptions of the data much more quickly. Figure 4B and C plot acquisition rate and mean accuracy with 5K search steps per trial, just 1% of the previous budget. Fleet's acquisition rate sharply declines while MPL's is ≥84% of that seen for the large budget. MPL is also reliably closer to human accuracy per function via a two-tailed paired sample Wilcoxon signed-rank test ($V = 874$, $p < 0.001$, effect size $= 0.39$, 95% CI $= [0.176, 0.634]$). MPL remains a good model participant for this task (Supplementary Note 4); Supplementary Note 5 contains more details on the errors individual models make and on correlations in accuracy between models.

While MPL (5K) performs well, 5000 search steps may approach humans' upper limit on this task. The median human response time is 14.7s, and the 75th percentile is 29.5s. If people respond slowly and search exceptionally quickly, say on the order of 5–10ms per step (e.g. by considering hypotheses in parallel or using very shallow networks of neurons[84]), they may take on the order of 3000–6000 steps. If a single step takes 500–1000ms, however, people may respond on the basis of just 30–60 steps, extremely few for a search-based program learning model. Though worse than MPL (5K), learning rates for MPL (500), MPL (50), and even MPL (20) still fall within the band of human performance (Fig. 5A). After just 5 trials at 10 steps/trial (i.e. 50 total search steps), MPL surpasses Metagol's, Enumerate's, and RobustFill's performance (Fig. 3A) and Fleet (5K)'s performance (Fig. 4C), all of which consumed orders of magnitude more search (see also Supplementary Note 5).

MPL leverages the idea that an inferential process, or metaprogram, can be simpler than the program it produces. If so, the probability of sampling a metaprogram should generally be higher than the probability of directly sampling the associated program, which would help explain MPL's high performance compared to alternative models. We find that 82.8% of metaprograms are at least as simple as their corresponding program (Fig. 5B; see also Supplementary Note 6).

MPL searches over metaprograms rather than over programs, but its prior (Eq. (12) in "Methods") is sensitive to both metaprogram complexity (i.e. cost of inferring a program) and program complexity (i.e. cost of representing a program). Both components are necessary—lesions sensitive to just one of the two components dramatically underperform the full model (Fig. 5C). The program prior encourages generalization and discourages memorization. The metaprogram prior may help MPL assign credit to useful metaprimitives and so search more efficiently.

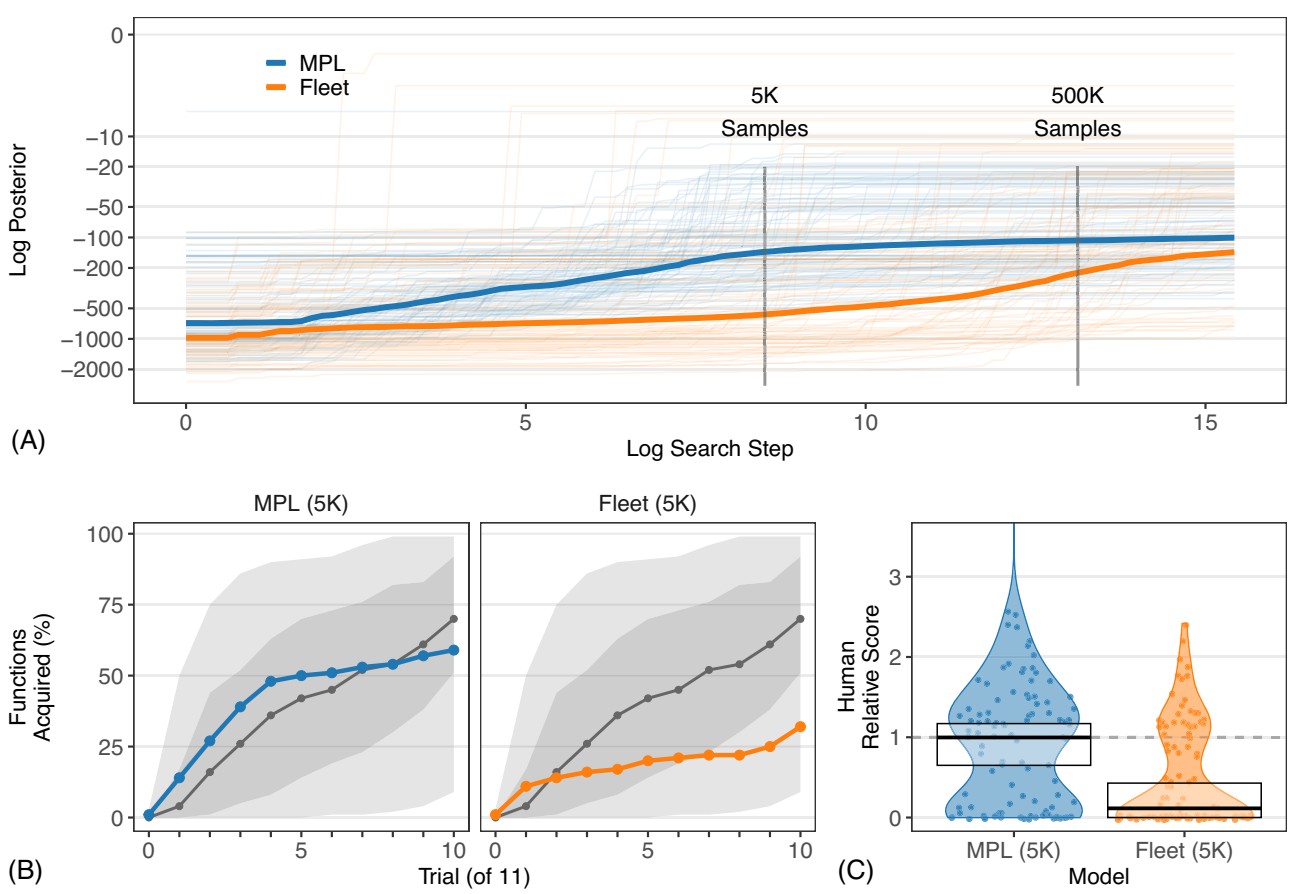

(A)

(B)

(C)

**Fig. 4 | MPL searches more efficiently than other models. A** Log$_e$ posterior of the best solution discovered by a given log$_e$ search step per function ($n = 100$ functions; thick = mean) per model with a fixed training set of 10 input/output examples per function. (**B**) and (**C**) follow Fig. 3A, B, respectively, with 5K search steps per trial.

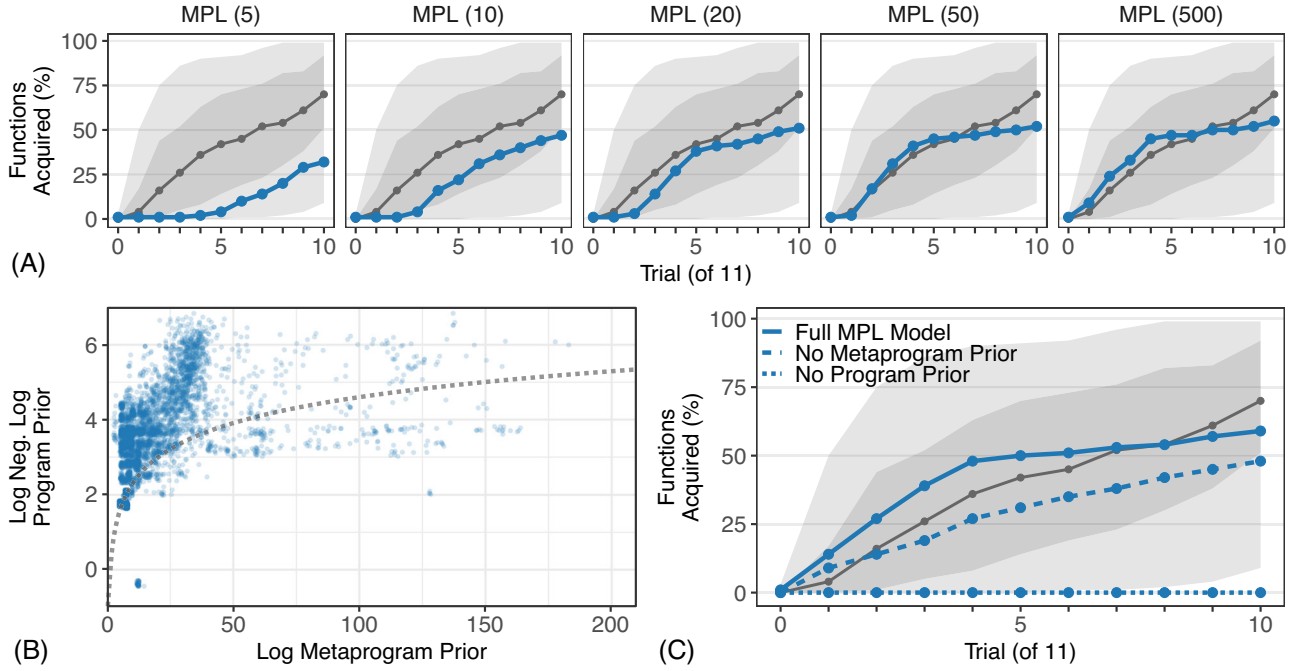

(A)

(B)

(C)

**Fig. 5 | Metaprimitives are central to MPL's performance. A** Follows (Fig. 3A), varying MPL's search steps per trial. **B** MPL's log$_e$ − log$_e$ program prior ($p_{\mathcal{P}}(\tilde{H})$) relative to MPL's log$_e$ metaprogram prior ($p_{\mathcal{M}}(H)$) for the highest-posterior hypotheses in each trial (dots; $n = 1,100$ trials) with parity between the two priors (curve). **C** Follows (Fig. 3A) for the full MPL model and when lesioning the two priors.

## Discussion

This paper uses functions over lists of natural numbers to test the hypothesis that people efficiently learn program-like representations by composing object-level operators and structured program transformations called metaprimitives. Instead of explaining learning purely in terms of the complexity of object-level content[5,49,76], this approach also incorporates the reasoning by which content is produced. An implementation of this theory, called MPL, uniquely achieves human-level performance in the test domain while capturing the hallmarks of human learning we emphasize in this paper: interpretable hypotheses; data efficiency; and computational efficiency. MPL does so by: (1) explicitly representing program transformations in the modeling language rather than merely implicitly in the search algorithm; (2) incorporating many kinds of program transformation rather than just one; and (3) extracting latent structure directly from data rather than discovering it by chance. Even so, MPL is only a first step toward more human-like models; we do not examine other essential traits like neural plausibility or the ability to generalize straightforwardly to related tasks.

These results reveal nuance in the relationship between simplicity and learning. All else being equal, people often prefer simpler explanations[85,86] and find them easier to acquire[5]. Classic program learning models thus strongly link psychological complexity to object-level simplicity. However, simplicity is language-dependent[87]—different primitives affect a language's inductive bias and thus how well it explains learning[49,71]. Relatedly, different axiomatic systems can produce shortest proofs of dramatically different lengths for the same theorem[88]. MPL's metaprimitives suggest a way to assess simplicity that goes beyond object-level content to incorporate structured inferences. These inferences reshape inductive bias, describing certain concepts easily but being poorly suited to others. Metaprograms are often shorter than programs because they can describe concepts in terms of observed data, which already contain relevant structure. Models tracking the complexity of both metaprograms and programs explain human learning better than models tracking just one or the other, suggesting that learning is sensitive to multiple kinds of simplicity.

Unless otherwise noted, all the models reported here use the same primitives as MPL and search over the same set of programs. We used a deliberately minimal DSL that could be easily implemented on a wide variety of models. For example, we do not include any higher-order functions in the DSL because many models, including Fleet, lack the typesystem needed to easily implement these functions. The key point here is that any program MPL discovered could also have been discovered by the other models, including Fleet.

What differentiates MPL is its use of metaprimitives, though it is important to note that MPL's success depends on having specific metaprimitives (and it might be possible to add metaprimitives that harm performance). A small collection of metaprimitives dramatically reshapes the initial inductive bias given by our expressive set of object-level primitives. For the problems studied here, this change in the inductive bias significantly improves the ability to explain human performance. Different primitives would almost certainly produce different results (e.g. performance would likely be much higher for all models if we added the target functions as primitives, or even if we moved from the primitives in Table 1 to those in Supplementary Table 2). More rigorously comparing a variety of languages with different combinations of primitives and metaprimitives—as has been done previously for primitives alone[49]—is a valuable direction for future work.

Metaprimitives seem likely to remain useful, however, because they can be sensitive to the internal structure of their arguments in ways that object-level primitives cannot. This sensitivity can allow metaprimitives to effectively prune the search space by ignoring hypotheses which are syntactically valid but inconsistent with the internal structure of their arguments. When search starts by observing or memorizing data—which already contains the structure to be explained—this pruning effect can sometimes allow search to quickly compose metaprimitives that reason backward from the data to a concise generating program. This approach overcomes shortcomings of traditional hypothesis-driven learners (which must discover relevant structure largely by chance) and data-driven learners (which typically apply a fixed pattern of reasoning).

We are not suggesting that it is only possible to encode the right inductive bias for a particular task using metaprimitives, but rather that metaprimitives provide a valuable and flexible way to encode a range of human-like inductive biases which rule-learning models can easily leverage. Some metaprimitives, like `AntiUnify`, are very general. A model would require many additional primitives and architectural changes to compensate for its loss. Others, such as our limited `Recurse` operator, might only require a couple of primitives or a change to the typesystem. More generally, metaprimitives are likely to excel when some pattern in a program's syntactic structure justifies transforming that program in a well-specified way. Primitives are likely to excel most when the internal structure of the arguments is largely irrelevant to the search process.

The diverse algorithms in our model comparison demonstrate that there are many ways to leverage composition, e.g. modifying sub-trees and using the rules of composition to constrain search. Future work can more systematically characterize the various ways composition can inform search and how each affects performance. Even more generally, it would be useful to precisely characterize the implications of adopting a compositional versus a non-compositional representation.

This paper demonstrates the promise of metaprimitives with an implemented example in the computationally universal list functions domain. Yet, neither program induction broadly nor the specific techniques we introduce here are limited to list functions. We focus on a benchmark of 100 problems emphasizing the modestly diverse set of computational patterns which MPL is capable of leveraging during search; this makes it possible to test our hypothesis by comparing solutions described with and without metaprimitives. Future metaprimitive models should address a broader set of problems by formalizing additional inference techniques and linking them to human behavior. This could include more sophisticated versions of the metaprimitives studied here, such as one capturing a more general set of fold-like computations or one capturing recursion with latent state. In addition, while `Memorize` and `AntiUnify` capture general patterns of reasoning, `Recurse` and `Compose` focus on transformations that are most useful only for limited classes of list functions. Metaprimitives are thus neither exclusively domain-specific nor domain-general, and their use could be extended to explicitly incorporate domain-specific analyses modeling well-known knowledge effects[64]. Developing a general model of the many forms of computational reasoning people can perform is likely to be a large-scale collaborative endeavor involving many kinds of empirical and computational experiments. What we aimed to do here was to take an initial and necessarily limited step toward such a model. We would not be surprised to find that humans use a much larger set of more sophisticated reasoning techniques than MPL. We would be surprised, however, to find that humans do not flexibly combine techniques for reasoning about data to significantly improve the speed of learning.

Future models can move beyond small and unchanging model languages to better match people's immense and largely learned cognitive repertoire[45,63]. Algorithms that expand modeling languages over time[71] begin to capture this dynamic, but more is needed. It remains unclear, for example, how to model people's apparent creation of genuinely novel symbols[89]. Finally, children go beyond collecting primitives; they appropriately select between them and can explain their choices[90]. MPL's stochastic search could be extended to

behave similarly by including additional elements of analytical synthesis[53,91,92] and pattern-based reasoning[93–95]. This work would help refine program learning into a comprehensive formal account of distinctively human learning.

## Methods
### List functions
We manually created a benchmark set of 250 list functions designed to vary widely in learning difficulty and algorithmic content. Each function can be expressed in a rich domain-specific language (DSL) embedded in a typed lambda calculus. Lambda calculus is a Turing-universal formalism that models computation as function abstraction and application[96]. It plays a fundamental role in computer science and frequently appears in computational models of learning[97–99]. We equip our language with a Hindley-Milner typesystem[100] which provides syntactic guarantees on the semantic correctness of programs. Intuitively, the type system eliminates programs which are semantically nonsensical (e.g. take the second element of the number 3) while allowing all semantically meaningful programs. Supplementary Table 1 describes the type system and Supplementary Table 2 describes the language primitives.

Supplementary Table 11 lists the 250 list functions in our dataset. 84 functions exclusively use the numbers 0–9; the remainder also use 10–99. The model comparison involved concepts c001–c100. Very few of these functions require numerical abilities beyond counting and basic arithmetic. The functions more typically focus on structural manipulations like inserting, swapping, or removing elements. The full 250-function dataset is intended as a benchmark for assessing human learners and future formal theories of learning; the language used to generate them contains many more primitives than the language available to model learners, which is described in the main text. The first 100 functions can be expressed in this much smaller language, making them more amenable to formal analysis by existing computational models. This 100-function subset still varies widely in terms of human learning and the algorithmic abilities required to express them, which include conditional, recursive, arithmetic, and pattern-based reasoning.

To generate input/output pairs for each function, we randomly generated one million sets of 11 input/output pairs and selected the best according to a per-function custom scoring function. Input and outputs were restricted to contain 0 to 15 elements. The per-function scoring function always favored variance in input and output length, variance in the elements of the lists, a high number of unique outputs, and a low number of examples in which the input and output were identical. Each was then also customized to favor features relevant to the given concept. For example, a concept indexing the third element might favor inputs with three or more elements, while a concept using the first element as an index might favor lists in which the first element was less than or equal to the length of the list. After selecting a set of examples, we then generated five thousand random orderings and selected the one with the highest score based on: applying the per-concept scoring function to the first five pairs, applying the per-concept scoring function to the last six pairs, whether the input differed from the output in the first example, and the distance between 5 and the length of the first input.

### Experimental procedure
We report the results of a behavioral experiment involving human participants. Our procedure complies with all relevant ethical regulations and was approved by the Institutional Review Board at Massachusetts Institute of Technology where the study was conducted. Participants provided informed consent and received a flat fee of $7.50 for participating plus a $0.01 bonus for each correct response. This study was not preregistered.

Supplementary Fig. 1 shows a representative display from the behavioral paradigm. Participants agreed to play a guessing game with the computer and began by reviewing the game's instructions. After a short comprehension check, participants completed 110 trials—10 rounds of 11 trials each, with the current round clearly indicated onscreen. In each round, the computer selected one of the 250 list functions as a rule for transforming input lists into output lists. Functions were selected uniformly at random for each participant; neither the experimenter nor the participant knew the functions being tested at the time of the experiment. Each function took a list of natural numbers as input and returned a list of natural numbers as output. Lists could include the numbers 0–99 as elements and contain 0–15 elements. To help participants learn the rule, the computer presented a series of trials. To begin each trial, the computer would show a novel input list and ask the participant to predict the output associated with the input by typing their predicted response into the text box. Participants were told that their job was to guess the rule and use it to correctly respond to as many of the computer's queries as possible. Participants were required to type in the entire list and had to do so without typos for their response to be considered correct. After each prediction, the computer revealed the correct output, ending the trial. The input, output, and participant prediction remained on screen for the rest of the experiment to reduce working memory load; participants could review it on any future trial, including those in subsequent rounds. The paradigm thus encouraged online learning in an attempt to reduce long-term memory demand and more accurately measure trial-by-trial generalization[49]. Progress indicators at the bottom of the screen informed participants of their performance and the number of remaining trials. At the end of each round, the computer asked participants to enter a natural language description of the rule they thought the computer had been using. The experiment ended with a brief demographical survey. No statistical methods were used to predetermine sample sizes but our sample sizes are similar to those reported in previous publications[49].

### Participants
In total, 498 people provided informed consent and participated in the experiment, hosted on Amazon Mechanical Turk using PsiTurk (https://psiturk.org). While we attempted to define highly learnable concepts, not all our participants appeared to make a good faith effort. This situation is typical for online experiments. Based on pilot data, we excluded participants who completed the experiment: in less than 20min; with fewer than 10 correct responses; or by giving the same response for more than 20 trials. This excluded 106 participants, a significant proportion of our original sample, raising concerns that the task was simply too difficult, perhaps due to its abstract formulation. Among the excluded participants, mean task time was 51.7 min (95% CI [46.1, 57.8]), number of mean correct responses was 10.2 (95% CI [7.6, 13.1]), and mean number of appearances of the most common response was 20.3 (95% CI [17.3, 23.7]). Only 4 of the 106 excluded participants mentioned task difficulty in their post-experiment survey. By contrast, 72 provided some sort of positive comment about liking the task or finding it engaging. To reinforce the trustworthiness of our findings, we conducted a targeted replication focused on the 100 functions in the model comparison (Supplementary Note 8). To increase participant engagement and data quality[101], we recruited participants through Prolific (https://prolific.co) rather than Amazon Mechanical Turk and, per Prolific's policies, provided compensation based on median time requirements. Critically, we excluded only a single replication participant using our original exclusion criteria and find results similar to our original sample (mean accuracy was actually significantly higher in the replication sample.). Together, these results show that the task is neither too abstract nor too difficult for participants. They instead suggest that, rather than excluding the low end of a single statistical distribution, the exclusion criteria separate an small

but expected group of participants failing to make a good faith effort from a much larger distribution of earnest participants.

We analyzed data from the remaining 392, where mean task time was 78.3min (95% CI [75.1, 81.9]), number of mean correct responses was 50.9 (95% CI [49.2, 52.7]), and mean number of appearances of the most common response was 6.0 (95% CI [5.6, 6.3]). Participant age for this group ranged from 18.6yrs to 69.4yrs (median: 39.2yrs), with 253 males, 132 females, and 2 of other genders (self-reported; 5 did not respond). Neither sex nor gender were included in the study design and did not figure into any reported analyses. We did not actively assess language skills but requested that participants speak English fluently. Participants received a median compensation of $8.00 for a median 72min of work. Participants found the task difficult but engaging with a mean self-reported difficulty rating of 4.9 and a mean self-reported engagement rating of 5.9, both on a 7-point Likert scale. Because each participant completed 10 rounds of trials, we collected data from about 16 participants for each list function. 3 of our pool of 392 participants were randomly assigned only functions that we do not analyze in this paper; this paper analyzes results from the remaining 389.

### Model procedure
Every model completed 5 runs of all 11 trials for each of the first 100 list functions in our dataset. As with people, learning progressed in an online fashion. For each trial $1 \le i \le 11$, the correct input/output pairs for the previous $i - 1$ trials were made available as training data, as well as the input for trial $i$. The correct output of trial $i$ was held out as test data. The training set was thus empty during the first trial, as it was for human participants. Each model except Metagol started trial $i + 1$ where trial $i$ finished, reusing computation from trials $1...i$ to hotstart trial $i + 1$. Metagol's design makes online learning difficult, so it treated trials independently. At the end of trial $i$'s search period, each model selected a best hypothesis and used it to predict an output for the current input. Each model used a similar DSL (i.e. the primitives in Table 1) with slight modifications to accommodate each model's particular representation format (e.g. lambda calculus, Prolog, term rewriting).

In abundant resource simulations, MPL and Fleet completed 500,000 search steps per trial (5,500,000 total) and the other models searched for 10min/trial. The larger budget allotted these other models allowed Enumerate to take more than one million steps per trial and Metagol to take more than one billion steps per trial. RobustFill took approximately 10,000 steps/trial but also benefited from amortizing inference over the course of three additional days spent training the neural network. In constrained resource simulations, MPL and Fleet completed 5000 search steps per trial unless otherwise clearly indicated. In the batch simulations (Fig. 4A), both MPL and Fleet completed five runs on each analyzed function. For each run, they observed the first ten of the eleven input/output pairs available for the target function and completed five million search steps.

### Comparison models
**Enumeration.** Enumerate[71] uses an exhaustive and symbolic technique known as enumerative search. It considers hypotheses approximately in order of description length, returning the first one consistent with observed data. This approach may seem implausible, but it tightly couples learning to simplicity measures like description length, as do humans in some domains[5]. It is also the simplest algorithm in this comparison and can be performed extremely quickly.

We used the high-performance enumeration algorithm from DreamCoder[71]. This model performs type-directed top-down grammar-based enumeration in approximately decreasing order of prior probability. That is, it treats the type system as a grammar over programs and, starting from a requested type, iteratively lists all programs matching the given type, starting with the shortest. The enumeration

proceeds in depth-first fashion, with an outer loop of iterative deepening: it first enumerates programs whose description length lies in $0-\Delta$, then all programs whose description length is $\Delta-2\Delta$, then $2\Delta-3\Delta$, and so on until the end of the trial. $\Delta$ was set to 1.5 nats; each task used a single CPU with no offline training or parameter learning. To accommodate online learning, Enumeration used a simple win-stay, lose-shift strategy[102]. When asked to make a prediction, it used the first program discovered which correctly explained all previously observed input/output pairs. If its predicted output was also correct, it continued to use that program to make predictions on subsequent trials. If the predicted output was incorrect, it would select the first program to correctly explain all previously observed input/output pairs plus the newly observed pair revealed after making the prediction. Assuming terminating programs, grammar-based enumeration is also guaranteed to discover the simplest possible solution[103] (Levin search[104] performs similarly with non-terminating programs).

**Stochastic search.** Fleet[42] is stochastic and symbolic. It samples from a Bayesian posterior over programs that balances simplicity against fit to data, consistent with psychological theories of learning as stochastic search[105]. This approach explains human learning in domains like Boolean concepts[49], counting routines[31], and kinship systems[106]. It is a forerunner of MPL but lacks metaprimitives in the language and a sensitivity to metaprograms in the prior.

Because exact sampling is intractable, Fleet uses a high performance implementation of the Rational Rules algorithm[76] for MCMC over programs. This technique proposes changes to entire subtrees of a program tree by selecting a node uniformly at random and regenerating it from the grammar. Our model also used a parallel tempering scheme[107] with five chains adaptively spaced to have efficient proposal acceptance rates. The maximum temperature was set to the trial number plus one, and the minimum temperature was fixed to 1.0, meaning the lowest temperature chain theoretically sampled from the target posterior. Swaps between chains were proposed every second and temperatures were adapted every 30s. The Fleet grammar did not include lambda abstraction due to limitations of the current implementation. Fleet is explicitly Bayesian. In these simulations, it used a grammar-based prior and a likelihood based on string edit distance (treating lists as strings of characters) which deleted each character from the end of a list with probability $10^{-4}$, and then appended uniformly random characters with the same probability. To support online learning, each new trial was started on the hypothesis with the best posterior in the preceding trial.

**Proof-driven search.** We used Metagol[54,81,108], an ILP system which uses a Prolog meta-interpreter to induce Prolog programs. Like Enumerate, Metagol is also exhaustive and symbolic but models learning as constraint satisfaction. It learns by recursively constructing a compact first-order logical proof which includes encodings of the data and task constraints. It builds on techniques which learn programs using Boolean formulae[109] or first-order clauses[72]. This approach aggressively prunes hypotheses known to be inconsistent with the data and learns successfully in many domains, including data transformation tasks similar to list functions[110]. It has also been used to model the way that humans' inductive bias shifts with repeated exposure to a domain[111].

Metagol uses metarules, or program templates, to restrict the form proofs can take. Metarules are higher-order clauses such that the goal of Metagol is to find substitutions for the higher-order variables. Deciding which metarules to use for a given task is an unsolved problem[112,113]. Supplementary Table 3 shows the eight metarules used by the Metagol simulations in this work. Metagol also induces longer clauses though predicate invention, similar to the introduction of lambda abstractions. Metagol works by partially constructing and evaluating programs, pruning the search space when a partial program fails to cover the positive examples or erroneously covers negative

examples. We only used positive examples in these simulations. Prolog programs encode nondeterministic relations. To evaluate Metagol, we called the learned Prolog program with the input given as the first argument and asked for answer substitutions for the second argument, taking the first provided substitution as the output.

**Neural program synthesis.** We used RobustFill[82], a stochastic algorithm that blends elements of neural and symbolic approaches to learning. It searches stochastically for programs guided by a deep neural network (in particular, a neural sequence-to-sequence encoder-decoder model with attention). Like Fleet, the network can be seen as approximating Bayesian inference over programs. RobustFill, however, uses a different technique for sampling programs. It samples a series of program symbols using weights generated by the network given observed data and the previous program symbol as input. It seems unlikely that human learning is either purely continuous or purely symbolic. We test RobustFill because it is neurosymbolic and because it outperformed both purely symbolic and purely neural approaches on string manipulation tasks similar to list functions.

Our implementation is nearly identical to the Attn-A RobustFill model[82]. The model differs in that we added a learned grammar mask using a separate LSTM language model over the program syntax[114]. The output probabilities of this LSTM were used to mask the output probabilities of the Robustfill model, encouraging the model to put less probability mass on grammatically invalid sequences. The model uses standard supervised, teacher-forcing techniques for training sequence to sequence models, minimizing cross-entropy loss on the training data. We used a hidden size of 512 and an embedding size of 128. We trained the network for 3 days. This meant approximately 105,000 iterations with a batchsize of 16 programs ( -1.6 million random programs seen during training). Training programs could have a maximum depth of 6, and each was associated with 1 to 10 input/output pairs, with the number of examples being sampled uniformly at random for each program.

**Large language models.** We used Codex[83], a stochastic neural model similar in spirit to RobustFill but which uses a different architecture trained on a far broader and bigger dataset. It is based on the large language model GPT-3[115], trained on hundreds of billions of tokens of text scraped from the Internet and fine-tuned on billions of lines of code from GitHub (https://github.com). We evaluate it here because its recent successes on reasoning and computer programming tasks suggest it as one of the most compelling models of intelligence available today.

We used the OpenAI API to run the Codex[83] model. Each task was presented in a few-shot manner, presenting four preceding example tasks with five input/output pairs each taken from the instructions in the human behavioral paradigm before presenting the test task. Each trial was completed independently; trial $n$ presented $n − 1$ complete input/output pairs followed by input $n$. To test Codex as a form of symbolic search, we asked it to produce python programs predicting outputs given inputs. The task embedded training data in a python docstring, requesting the body of a python function that would produce the corresponding output when applied to the test input. API calls requested a single response at temperature 0 and ended at the first newline or after a maximum of 150 tokens, whichever came first. Because it is unclear whether GPT-3 or Codex had access to our original benchmark data, which is publicly available online, we also generated novel input/output pairs to test Codex. Performance was similar to using the original stimuli from the human behavioral experiment, so we report results using the original stimuli.

## MPL (MetaProgram Learner) model
MPL represents programs as first-order term rewriting systems (TRS)[116,117] (Supplementary Note 1). They are a less common basis for program synthesis systems than alternative representations like first-order logic[72,118], combinatory logic[13,119,120] or lambda calculus[38,49,71,121], but have previously appeared in inductive learning systems[122,123]. MPL augments a user-provided domain-specific language consisting of object-level primitives with a set of metaprimitives (Supplementary Note 2).

To balance simplicity and fit, MPL models learning as MAP inference in a Bayesian posterior over metaprograms computed using Bayes' Law:

$$p(H \mid D) \propto p(D \mid H)p(H). \tag{11}$$

where $H$ is a metaprogram reducing to program $\widetilde{H}$ given data, $D$. $p(H)$ is given as

$$p(H) \propto \exp\left(\frac{\ln p_{\mathcal{M}}(H) + \ln p_{\mathcal{P}}(\widetilde{H})}{2}\right) \tag{12}$$

where $p_{\mathcal{M}}$ is a grammar-based metaprogram prior given by a type-constrained probabilistic context-sensitive grammar over primitives and metaprimitives and $p_{\mathcal{P}}$ is a similar grammar-based program prior over just primitives. Both favor simple expressions.

MPL assumes that each input/output pair, $(x, y)$, is generated independently. $p(D \mid H)$ is a prefix-based likelihood[42] It scores responses by assuming a noise process that deletes from and appends to lists stochastically such that each change occurs with probability $\eta$ (In all our experiments, $\eta = 10^{-6}$.). Output likelihood increases with the size of its common prefix with the correct response. If append operations can select from $N$ characters, $\widetilde{H}(x)$ is the predicted output for input $x$ using metaprogram $H$, $\mathbb{I}[x,y,i]$ indicates whether lists $x$ and $y$ share a prefix of length $i$, and $|x|$ is the length of list $x$, then:

$$p(D \mid H) = \sum_{(x,y)\in D} \sum_{i=0}^{\min(|\widetilde{H}(x)|,|y|)} \mathbb{I}[\widetilde{H}(x),y,i]\eta^{|\widetilde{H}(x)|-i}\left(\frac{\eta}{N}\right)^{|y|-i}(1-\eta)^{1+\min(i,1)} \tag{13}$$

This likelihood is useful whenever the output contains multiple elements that can be explained incrementally. This is the case both for recursive functions producing multiple elements (e.g. remove every other element), but it is also useful for non-recursive problems such as $\mathcal{G}$ (Eq. (10)). The prefix-bias will be less helpful for functions which recursively fold or reduce the input into a single element (e.g. input length) and for functions which select a single element of the input non-recursively (e.g. the third input element).

Computing the posterior exactly is intractable; MPL approximates it using Markov Chain Monte Carlo (MCMC) over programs[42,76] extended to the space of metaprograms. Inference used a custom implementation of parallel tempering with two pools of five temperatures each, ranging from 1.0 to the current trial number plus one, spaced exponentially, and proposing swaps every 25s. One pool searched over hypotheses formed from the full DSL, i.e. the object-level DSL plus the MPL metaprimitives. The other used the object-level DSL only, i.e. primitives only. Chains used tree-regeneration proposals[76] and custom proposals for inserting, removing, and regenerating metaprimitives.

The model had access to instances of both pools, which maintained separate state but reported their hypotheses to a shared collection of the best hypotheses observed by either pool. At each search step, the model would collect a single sample from either pool but not both. This decision was made randomly, choosing the full DSL pool with probability $\alpha$ and the object-level DSL pool with probability $1 − \alpha$. The auxiliary model varies $\alpha$, while all other experiments fix it to 1.0.

MPL considered metaprograms containing 50 or fewer random choices and 7 or fewer metaprimitives. It also only considered

metaprograms producing deterministic TRSs. To support online learning, MPL retained paths to the 100 top-scoring solutions between trials and initialized chains for the next trial using the best known hypothesis.

## Reporting summary

Further information on research design is available in the Nature Portfolio Reporting Summary linked to this article.

## Data availability

The raw model data and processed human data generated in this study have been deposited in the Open Science Foundation database at https://doi.org/10.17605/OSF.IO/GQ2HJ. The raw human data generated in this study are protected and are not available due to data privacy laws.

## Code availability

The code needed to reproduce the figures and results has been deposited in the Open Science Foundation database at https://doi.org/10.17605/OSF.IO/GQ2HJ. Key libraries are also available for: Markov Chain Monte Carlo over programs (https://github.com/joshrule/program-induction); term rewriting systems (https://github.com/joshrule/term-rewriting-rs); and Hindley-Milner type inference (https://github.com/joshrule/polytype-rs).

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

## Acknowledgements

This work was largely conducted while J.S.R. was a graduate student at Massachusetts Institute of Technology. It was supported by National Science Foundation (NSF), Division of Research on Learning Grant 1760874 (S.T.P.), Eunice Kennedy Shriver National Institute of Child Health & Human Development at the National Institutes of Health Award 1R01HD085996 (S.T.P.), NSF Graduate Research Fellowship Grants 1122374 & 1745302 (J.S.R.), Office of Naval Research Grant N00014-18-1-2847 (J.B.T.), NSF STC Award CCF-1231216 for the Center for Minds, Brains and Machines (J.B.T.), Air Force Office of Scientific Research Award FA9550-19-1-0269 (J.B.T.), and the Siegel Family Endowment (J.B.T.).

## Author contributions

J.S.R., S.T.P., and J.B.T. conceived the study and method. J.S.R., S.T.P., A.C., K.E., and M.N. developed software and ran simulations. J.S.R. and S.T.P. analyzed and visualized the data. J.S.R., S.T.P. and J.B.T. drafted the paper, and all authors revised the paper. S.T.P. and J.B.T. contributed resources and supervision.

## Competing interests

The authors declare no competing interests.
