## [Peer Review File - NEW · Nature Communications]

Symbolic metaprogram search improves learning efficiency and explains rule learning in humansEditorial Note: This manuscript has been previously reviewed at another journal that is not operating a transparent peer review scheme. This document only contains reviewer comments and rebuttal letters for versions considered at *Nature Communications*. Mentions of the other journal have been redacted.

Reviewer #1 (Remarks to the Author):

I found this paper impressive when it was considered for [REDACTED], and I still find it impressive. I think the rewriting has improved the focus, and I think the novel conclusion is even more clear: the use of metaprogram search substantially improves learning, and more closely models

human performance on the kind of learning problems the authors consider.

My only hint of concern about the overall conclusion, already discussed in the previous review cycle, is that while the types of primitives the authors consider are effective choices for the type of learning problem (numeric sequences) considered, they might not be for more naturalistic learning problems. However in the new draft I think the choice of problem is appropriately motivated, and the conclusions are not overstated, so I no longer think this is a substantial concern.

For these reasons, I (again) recommend publication.

My remaining points are typos and minor issues to improve the paper.

First, I would urge the authors to choose a different abbreviation for Hacker-Like than HL. In many contexts HL stands for "human learning", as opposed to ML (machine learning). Perhaps just use "Hacker" which is even more memorable.

Line 8 - dramatically

Line 39 - LOT has not yet been introduced or explained.

Line 43 - "Humans, like computer programmers, fluidly operate..." implies that computer programmers are not humans. Please rephrase.

Line 45 - "Symbolic programs provide interpretable hypotheses by decomposing into discrete and semantically meaningful parts that support modular explanation, reuse, and sharing"
Decomposing WHAT into ...?

Line 69 - In an earlier version of the paper, I urged to authors to clarify where their model sits in terms of "levels of explanation". For what it's worth, I think the new verbiage about this topic is quite reasonable and deftly handles the issue. It's somewhere in between, fair enough.

Line 380 - A little peculiar to have a 2020 paper cited for this very old point.

Line 468 - "inconsistent with the internal structure of its arguments." Who is "it" in this sentence? I think the authors mean "their".

Signed,

Reviewer #2 (Remarks to the Author):

Thank you for the opportunity to re-review this manuscript. I have carefully read both the revision and response letter with great interest and found that the authors have thoroughly and thoughtfully addressed the concerns that I raised in the two earlier rounds of review at another journal. In particular, I find the more targeted framing in the Introduction to be quite effective at setting up the studies reported in this paper. In addition, the extra methodological detail concerning the behavioural studies is appreciated. I believe that this work will provide a significant contribution to the literature, and I look forward to the opportunity to cite it in the near future.

Here is an easily fixed typo: In the sentence: "They go beyond associative pairings or even simple logical or arithmetic formulae to encodes a series of steps..." I assume it should be "encode" rather than "encodes."

Reviewer #3 (Remarks to the Author):

Summary:

This paper introduces the concept of metaprogram search, a novel representation for rule induction. This paper takes the stance that human rule induction is analogous to program induction. Taking this analogy further, the authors claim

that recent advances in program synthesis could be used as an analogy to describe possible ways of thinking. However, this analogy breaks down when one looks at state-of-the-art approaches towards program induction, as they require very efficiently enumerating candidate programs symbolically, or they use neural networks to spit out high probability tokens. This paper claims that one representation for human thought is that of "metaprograms" or "metarules". In this model, candidate programs are processed, but so too are "metaprograms" -- part of thought is taking an existing program (or rule set) and transforming it in some way, this is the "HL" (Hacker-Like) Model.

This paper describes this approach and provides a formal language for one instantiation of the HL model on programs, and provide a search strategy through the space of HL programs. They build a benchmark of 100 challenges, and compare the HL model to Fleet, Enumerate, Metagol, RobustFill, Codex and humans. The authors claim that the HL model performs more analogously to humans than the other synthesis engines.

Thoughts:

I found this paper quite interesting, and I enjoyed thinking about how I personally address tasks. Intuitively, the paper makes sense, I have some aspect of "metaprograms" where I transform programs that I've thought of into edited alternative versions. Obviously this model is imperfect as I don't think many people's "rules" are so perfectly defined and small, nor do most people have such a small "language" for such rules.

There are some obvious gaps that make this not a full and complete description of human cognition. But at the same time... that is obviously true. Ultimately, this paper does convince me that it improves upon prior models for human cognition (note: I can't claim on whether it fully evaluates related work in the

domain, as I am not an expert).

The evaluation is relatively extensive, and I agree with the broad conclusions. I agree that HL seems to fit human cognition better. But there are some little things that are either overclaimed or stated without citations that I'm unconvinced by. For example, is it fair to say that Codex's training demands are implausible for human cognition? (Take the following portion with a grain of salt, I am not an expert in cognition) I would say that Codex's "runtime" is more similar to the hardware our brains run on than any form of symbolic description. And yes, maybe the mechanism that trains Codex is different, but the actual procedure that is being run I think is more similar than is described. Is this paper discussing how we train or how we think? While the training is different (large corpuses of data that is learned inefficiently from as opposed to relatively quick training that is enabled by very very long-scale training of evolution). I know I'm harping on one sentence, but it's a sentence that really frustrated me, if you are trying to discuss how humans solve tasks, focus on that.

Additionally, it doesn't really bring into account the huge context difference between the models and a human. Humans have an incredibly large context of possible rules. This model really could not remotely handle the large context of humans alone. Some form of gradient descent alongside metaprograms is simply unable to even approach humans capabilities of identifying core concepts that are needed. I am fairly certain the number of steps that would be taken to remotely get near human correctness would make this approach presume incredibly fast thinkers. (I also think 5-10ms to discard a candidate program does not fit at all with my intuition of how fast human thought is -- this number really needs to be justified).

However, I do find this paper interesting, and I just think this paper needs a bit of tempering at places, an additional discussion about how the task proposed

to the computers is (inherently, due to the fact we are not single-purpose)
different than the tasks given to the humans, particularly as it relates to the
space of concepts given to these synthesizers.

Nits:

l8: dramtically <- spelling

l21: rules face a number of challenges. <- awkward

l39: LOT <- Is this an acronym? If so, provide the full version too. If not, why capitalized

l226: Work on number <- awkward

Response to the reviewers

What follows are the comments from the reviewers, as well as our detailed response. Reviewer comments are marked in **bold**, our response in standard font, and quotations from the revised manuscript are given in block quotes, with changes marked in blue.

Reviewer #1 (Remarks to the Author):

I found this paper impressive when it was considered for [REDACTED], and I still find it impressive. I think the rewriting has improved the focus, and I think the novel conclusion is even more clear: the use of metaprogram search substantially improves learning, and more closely models human performance on the kind of learning problems the author consider.

My only hint of concern about the overall conclusion, already discussed in the previous review cycle, is that while the types of primitives the authors consider are effective choices for the type of learning problem (numeric sequences) considered, they might not be for more naturalistic learning problems. However in the new draft I think the choice of problem is appropriately motivated, and the conclusions are not overstated, so I no longer think this is a substantial concern.

For these reasons, I (again) recommend publication.

Thank you—we appreciate your kind words. We are grateful for your recommendation and are glad to hear that our revisions have addressed your primary concerns with earlier versions of the manuscript.

My remaining points are typos and minor issues to improve the paper.

First, I would urge the authors to choose a different abbreviation for Hacker-Like than HL. In many contexts HL stands for "human learning", as opposed to ML (machine learning). Perhaps just use "Hacker" which is even more memorable.

We appreciate this suggestion. To avoid confusion, we have renamed the Hacker-Like (HL) model and now refer to it throughout the paper as the MetaProgram Learner (MPL). While this makes the connection to hacking less obvious, it more clearly emphasizes the core insight of the model, which is to move from learning object-level programs to learning metaprograms.

Line 8 - dramatically

Thank you—fixed!

Here we show that symbolic search over the space of metaprograms—programs that revise programs—dramatically improves learning efficiency.

Line 39 - LOT has not yet been introduced or explained.

Thank you—we simply removed the use of the acronym.

One theory of rule-learning treats the language of thought as a sort of mental programming language, such that learning proceeds by constructing program-like representations.

Line 43 - "Humans, like computer programmers, fluidly operate..." implies that computer programmers are not humans. Please rephrase.

It is certainly not our intent to suggest that human computer programmers are not human. We have rephrased this sentence:

Humans learning new rules, much like computer programmers writing new programs, fluidly operate over a broad space of computations and appear to efficiently construct interpretable structures from sparse data.¹⁹

Line 45 - "Symbolic programs provide interpretable hypotheses by decomposing into discrete and semantically meaningful parts that support modular explanation, reuse, and sharing" Decomposing WHAT into ...?

Our original intention with this sentence was to communicate that the programs themselves decompose into simpler parts, namely the statements, expressions, and even the individual symbols from which they are composed. It is perhaps even clearer, however, to focus on how programs provide an interpretable way to understand the structure of complex *computations*, by showing how they can be decomposed into simpler computations. We have taken this approach in the revision:

Symbolic programs provide interpretable hypotheses by decomposing complex computations into discrete and semantically meaningful parts—i.e. simpler computations—that support modular explanation, reuse, and sharing.²⁹

Line 69 - In an earlier version of the paper, I urged to authors to clarify where their model sits in terms of "levels of explanation". For what it's worth, I think the new verbiage about this topic is quite reasonable and deftly handles the issue. It's somewhere in between, fair enough.

Thank you for noting your appreciation of this change. We wholeheartedly agree that being clear about which levels of explanation we intend to address with this work sharpens both our presentation and our own understanding.

Line 380 - A little peculiar to have a 2020 paper cited for this very old point.

While we agree that the idea that cognition is resource-constrained is old, it is one that continues to bear fruit. This 2020 paper reviews the history of the idea, summarizes recent work, and identifies key open questions. It thus strikes us as a good place for readers to begin familiarizing themselves with resource rationality / bounded rationality and provides many more references for further study than we could reasonably provide here.

Line 468 - "inconsistent with the internal structure of its arguments." Who is "it" in this sentence? I think the authors mean "their".

Thank you—fixed!

This sensitivity can allow metaprimatives to effectively prune the search space by ignoring hypotheses which are syntactically valid but inconsistent with the internal structure of their arguments.

Signed,

Reviewer 2

Thank you for the opportunity to re-review this manuscript. I have carefully read both the revision and response letter with great interest and found that the authors have thoroughly and thoughtfully addressed the concerns that I raised in the two earlier rounds of review at another journal. In particular, I find the more targeted framing in the Introduction to be quite effective at setting up the studies reported in this paper. In addition, the extra methodological detail concerning the behavioural studies is appreciated. I believe that this work will provide a significant contribution to the literature, and I look forward to the opportunity to cite it in the near future.

Thank you for these kind words and for recommending the paper for publication.

Here is an easily fixed typo: In the sentence: "They go beyond associative pairings or even simple logical or arithmetic formulae to encodes a series of steps..." I assume it should be "encode" rather than "encodes."

Thank you for noting this typo. We have fixed it in our revision:

They go beyond associative pairings or even simple logical or arithmetic formulae to encode a series of steps incorporating a variety of algorithmic content.¹⁹

Reviewer 3

Summary:

This paper introduces the concept of metaprogram search, a novel representation for rule induction. This paper takes the stance that human rule induction is analogous to program induction. Taking this analogy further, the authors claim that recent advances in program synthesis could be used as an analogy to describe possible ways of thinking. However, this analogy breaks down when one looks at state-of-the-art approaches towards program induction, as they require very efficiently enumerating candidate programs symbolically, or they use neural networks to spit out high probability tokens. This paper claims that one representation for human thought is that of "metaprograms" or "metarules". In this model, candidate programs are processed, but so too are "metaprograms" -- part of thought is taking an existing program (or rule set) and transforming it in some way, this is the "HL" (Hacker-Like) Model.

This paper describes this approach and provides a formal language for one instantiation of the HL model on programs, and provide a search strategy through the space of HL programs. They build a benchmark of 100 challenges, and compare the HL model to Fleet, Enumerate, Metagol, RobustFill, Codex and humans. The authors claim that the HL model performs more analogously to humans than the other synthesis engines.

Thank you—you have accurately summarized the key aspects of this work.

Thoughts:

I found this paper quite interesting, and I enjoyed thinking about how I personally address tasks. Intuitively, the paper makes sense, I have some aspect of "metaprograms" where I transform programs that I've thought of into edited alternative versions. Obviously this model is imperfect as I don't think many people's "rules" are so perfectly defined and small, nor do most people have such a small "language" for such rules.

There are some obvious gaps that make this not a full and complete description of human cognition. But at the same time... that is obviously true. Ultimately, this paper does convince me that it improves upon prior models for human cognition (note: I can't claim on whether it fully evaluates related work in the domain, as I am not an expert).

The evaluation is relatively extensive, and I agree with the broad conclusions. I agree that HL seems to fit human cognition better. But there are some little things that are either overclaimed or stated without citations that I'm

unconvinced by. For example, is it fair to say that Codex's training demands are implausible for human cognition? (Take the following portion with a grain of salt, I am not an expert in cognition) I would say that Codex's "runtime" is more similar to the hardware our brains run on than any form of symbolic description. And yes, maybe the mechanism that trains Codex is different, but the actual procedure that is being run I think is more similar than is described. Is this paper discussing how we train or how we think? While the training is different (large corpuses of data that is learned inefficiently from as opposed to relatively quick training that is enabled by very very long-scale training of evolution). I know I'm harping on one sentence, but it's a sentence that really frustrated me, if you are trying to discuss how humans solve tasks, focus on that.

Thank you for this helpful comment. We agree that the core issue at stake here is not how the models were developed (e.g. manual engineering vs. automatic training using large datasets). That question is interesting but best saved for another time. The key question here is which models best explain human performance on the particular learning tasks we test. We have therefore removed any discussion of Codex focused purely on how it was trained.

Codex⁸⁸ is a stochastic neural model similar in spirit to RobustFill but which uses a different architecture trained on a far broader and bigger dataset. It is based on the large language model GPT-3,⁸⁹ trained on hundreds of billions of tokens of text scraped from the Internet and fine-tuned on billions of lines of code from GitHub (<https://github.com>). We evaluate it here because its recent successes on reasoning and computer programming tasks suggest it as one of the most compelling models of intelligence available today.

RobustFill is neither exhaustive nor hillclimbing but generates independent samples (conditioned on the training data), which is extremely inefficient for low-probability programs. Codex also generates conditionally independent samples, but its significantly larger training set and more sophisticated architecture help it to outperform RobustFill.

We have also moved the discussion of participant programming experience, previously tied to a comment about Codex's extensive exposure to code during training, to a standalone paragraph:

Participants' performance is perhaps particularly impressive given their relatively low levels of programming experience. Of the 392 participants in our sample, 259 (66%) provided an interpretable free-response statement of their prior programming experience. Of these, 151 (58%; mean accuracy = 0.49) indicate no prior programming experience, an additional 27 (10%; mean accuracy = 0.50) indicate social exposure to programming concepts and perhaps simple website construction. 43 (17%; mean accuracy = 0.50) report encountering programming through introductory coursework or by building several websites. Only 38 (15%, mean accuracy = 0.53) indicate significant academic or professional exposure to programming (See also Supplementary Note 8).

Additionally, it doesn't really bring into account the huge context difference between the models and a human. Humans have an incredibly large context of possible rules. This model really could not remotely handle the large context of humans alone. Some form of gradient descent alongside metaprograms is simply unable to even approach humans capabilities of identifying core concepts that are needed. I am fairly certain the number of steps that would be taken to remotely get near human correctness would make this approach presume incredibly fast thinkers.

We agree that no existing computational model, including the model we introduce in this paper, fully captures the broad scope of human learning. This is especially true when we think about the relatively limited set of primitives available to most models. We have attempted to make this clear when introducing our approach:

Program-induction-based models of concept learning often use languages whose primitives (and in this case, metaprimitives) are closely related to the concepts being studied...The claim is not that these limited languages constitute a learner's entire mental repertoire, nor that the studied domain is the only one in which humans are capable of learning. Nor is the claim that the simple existence of computational primitives (or metaprimitives) is enough to explain human learning, or that any existing model is sufficient to explain all of human learning. They are instead case studies comparing a plausible set of primitives and learning dynamics against human learners in a particular domain. We take the same approach in introducing metaprimitives.

Later, in the Discussion, we discuss the need for a richer set of metaprimitives suited to both domain-general and domain-specific reasoning, as well as modeling approaches which grow and adapt the set of (meta)primitives over time, as humans' do:

Future metaprimitive models should address a broader set of problems by formalizing additional inference techniques and linking them to human behavior. This could include more sophisticated versions of the metaprimitives studied here, such as one capturing a more general set of fold-like computations or one capturing recursion with latent state. In addition, while Memorize and AntiUnify capture general patterns of reasoning, Recurse and Compose focus on transformations that are most useful only for limited classes of list functions. Metaprimitives are thus neither exclusively domain-specific nor domain-general, and their use could be extended to explicitly incorporate domain-specific analyses modeling well-known knowledge effects.⁶⁴ Developing a general model of the many forms of computational reasoning people can perform is likely to be a large-scale collaborative endeavor involving many kinds of empirical and computational experiments. What we aimed to do here was to take an initial and necessarily limited step toward such a model. We would not be surprised to find that humans use a much larger set of more sophisticated reasoning techniques than MPL. We would be surprised, however, to find that humans do not flexibly combine techniques for reasoning about data to significantly improve the speed of learning.

Future models can move beyond small and unchanging model languages to better match people's immense and largely learned cognitive repertoire.^{45,63} Algorithms that expand modeling languages over time⁷¹ begin to capture this dynamic, but more is needed. It remains unclear, for example, how to model people's apparent creation of genuinely novel symbols.⁹⁵ Finally, children go beyond collecting primitives; they appropriately select between them and can explain their choices.⁹⁶ MPL's stochastic search could be extended to behave similarly by including additional elements of analytical synthesis^{53,97,98} and pattern-based reasoning.⁹⁹⁻¹⁰¹ This work would help refine program learning into a comprehensive formal account of distinctively human learning.

(I also think 5-10ms to discard a candidate program does not fit at all with my intuition of how fast human thought is -- this number really needs to be justified).

Our intention here is to estimate a rough upper limit on the number of search steps which most people might perform on most problems. We justify participants taking approximately 30s/trial based on the 75th percentile timing observed in our human data. We justify the estimate of 5–10ms per hypothesis as occurring in a scenario where search considers multiple hypotheses in parallel or depends on a network of neurons that is just a few layers deep. These numbers are intended as merely a rough estimate to help show that the number of hypotheses human learners consider may be much smaller than the budget typically allocated to search in computational models.

If people respond slowly and search **exceptionally** quickly, say on the order of 5–10ms per step (e.g. by considering hypotheses in parallel or using very shallow networks of neurons⁹⁰), they may take on the order of 3,000–6,000 steps.

However, I do find this paper interesting, and I just think this paper needs a bit of tempering at places, an additional discussion about how the task proposed to the computers is (inherently, due to the fact we are not single-purpose) different than the tasks given to the humans, particularly as it relates to the space of concepts given to these synthesizers.

We are glad that you find the paper interesting and appreciate how your suggested changes have strengthened the paper.

Nits:

l8: dramatically <- spelling

Thank you—fixed!

Here we show that symbolic search over the space of metaprograms—programs that revise programs—**dramatically** improves learning efficiency.

I21: rules face a number of challenges. <- awkward

Thank you—we have rephrased to use a clearer and less awkward construction:

| The exact scope of human rule-learning is unclear: even if they *can* describe a wide variety of concepts,^{12,13} theories of rule-learning face a number of challenges.^{14–16}

I39: LOT <- Is this an acronym? If so, provide the full version too. If not, why capitalized

Thank you—LOT is an acronym for “Language of Thought”. Because we had not introduced it previously and only used it once in the manuscript, we simply substituted the full text.

| One theory of rule-learning treats the language of thought as a sort of mental programming language, such that learning proceeds by constructing program-like representations.

I226: Work on number <- awkward

Thank you—fixed!

| This can be seen, for example, in recent work on learning in the domains of number,^{31,75} logic,^{49,76} and geometry,^{37,77} among others.

Reviewer #3 (Remarks to the Author):

I am quite happy with the revisions provided. I appreciate that the authors were willing to temper some of their claims, and add in a few extra bits. I have no further issues with the paper, and recommend acceptance.